# Overburdened ferroptotic stress impairs tooth morphogenesis

**Haisheng Wang[1†], Xiaofeng Wang[1,2†], Liuyan Huang[1,2], Chenglin Wang[2], Fanyuan Yu[1,2]\*, Ling Ye[1,2]\***

[1]State Key Laboratory of Oral Diseases & National Center for Stomatology & National Clinical Research Center for Oral Diseases, West China Hospital of Stomatology, Sichuan University, Chengdu, China; [2]Department of Endodontics, West China School of Stomatology, Sichuan University, Chengdu, China

**Abstract** The role of regulated cell death in organ development, particularly the impact of non-apoptotic cell death, remains largely uncharted. Ferroptosis, a non-apoptotic cell death pathway known for its iron dependence and lethal lipid peroxidation, is currently being rigorously investigated for its pathological functions. The balance between ferroptotic stress (iron and iron-dependent lipid peroxidation) and ferroptosis supervising pathways (anti-lipid peroxidation systems) serves as the key mechanism regulating the activation of ferroptosis. Compared with other forms of regulated necrotic cell death, ferroptosis is critically related to the metabolism of lipid and iron which are also important in organ development. In our study, we examined the role of ferroptosis in organogenesis using an ex vivo tooth germ culture model, investigating the presence and impact of ferroptotic stress on tooth germ development. Our findings revealed that ferroptotic stress increased during tooth development, while the expression of glutathione peroxidase 4 (Gpx4), a crucial anti-lipid peroxidation enzyme, also escalated in dental epithelium/mesenchyme cells. The inhibition of ferroptosis was found to partially rescue erastin-impaired tooth morphogenesis. Our results suggest that while ferroptotic stress is present during tooth organogenesis, its effects are efficaciously controlled by the subsequent upregulation of Gpx4. Notably, an overabundance of ferroptotic stress, as induced by erastin, suppresses tooth morphogenesis.

## eLife assessment

This **important** study time elegantly demonstrates that ferroptotic stress may play critical roles in regulating tooth germ development. The evidence presented is **compelling**, based on an explant model and providing novel mechanistic insights into tooth development.

## Introduction

A delicate balance between cell division and regulated cell death (RCD) is crucial for the development of tissues and organs (**Ghose and Shaham, 2020**). In organs such as the nervous, immune, and reproductive systems, cells are initially produced in excess and are subsequently removed by RCD. During organ development, structures with transient functions are also eliminated by RCD when they are no longer necessary (**Reynaud and Driancourt, 2000**). The best-known example of this is the formation of digits in higher vertebrates, where the interdigital webs are eliminated primarily through the apoptotic machinery (**Lindsten et al., 2000**). RCD plays a vital role in animal development and adult life by eliminating abnormal and potentially harmful cells (**Opferman and Korsmeyer, 2003**). Apoptosis is the most extensively studied form of RCD which shrinks the nucleus and buds the plasma membrane

**\*For correspondence:**
fanyuan_yu@outlook.com (FY);
yeling@scu.edu.cn (LY)

[†]These authors contributed equally to this work

(without rupturing it), making it essential in achieving cell number regulation, tissue remodeling, and sculpting structures, driving morphogenesis during organ development (*Fuchs and Steller, 2011*).

However, several new types of cell death, including pyroptosis, necroptosis, and ferroptosis, have been discovered (*Galluzzi et al., 2018*). Unlike apoptosis, these newly discovered RCDs are characterized by pore formation and/or rupture of the plasma membrane, thus termed as regulated necrotic cell death (RNCD) (*Bedoui et al., 2020*). While apoptosis benefits organ morphogenesis in certain circumstances, whether RNCD and its regulatory mechanisms are biologically required in organ development remains poorly understood. Pyroptosis and necroptosis are mainly activated by external pathogens or damage-associated molecular patterns (*Yuan et al., 2016*), whereas ferroptosis is a distinct form of iron-dependent, lipid peroxidation-driven programmed cell death (*Stockwell, 2022*). Ferroptosis is a particular type of cell death that links lipid metabolism, reactive oxygen species (ROS) biology, iron regulation, and disease (*Stockwell et al., 2017*). Despite the pathological role of ferroptosis and/or ferroptotic stress in multiple diseases (*Zou and Schreiber, 2020*; *Jiang et al., 2021*), recent studies have identified the occurrence of ferroptosis in embryonal erythropoiesis and aging (*Zheng et al., 2021*) and aged skeletal muscle (*Ding et al., 2021*), indicating its unrecognized role in physiological processes. Unlike autonomous activation of apoptosis during organ development, the broken balance between ferroptotic stress and its suppressing mechanisms is now considered the main reason for ferroptosis. Recent studies demonstrated that MYSM1 deficiency causes human hematopoietic stem cell loss by ferroptosis, highlighting the broader developmental and regenerative role of ferroptosis (*Zhao et al., 2023*). The investigation into the potential involvement of ferroptosis and/or ferroptotic stress, other than rest forms of non-apoptotic cell death like pyroptosis and necroptosis, in organ morphogenesis is an area of great interest. However, conducting high-throughput analysis through in vivo studies presents challenges as it relies heavily on generating specific transgenic mice since erastin, the classic small molecule used to induce ferroptosis, is feasible in vivo (*Gao et al., 2016*). Thus, there is a crucial need for a more adequate model to explore the ferroptosis/ferroptotic stress involved in development.

The process of tooth development, which contains cell number regulation (proliferation, differentiation, and extinction of ameloblast), tissue remodeling (elimination of transient structure called enamel knot), and sculpting structures (cusp formation) (*Zhang et al., 2005*; *Liu et al., 2008*), is an ideal model for the investigation of RCD in organ development. Moreover, ex vivo culture of tooth germ is a well-established study system that allows the controlled manipulation of environmental, genetic, and pharmacological factors (like erastin), which can influence the developmental events of tooth (*He et al., 2019*), providing compelling evidence for its usefulness as an efficient research model in studying ferroptosis/ferroptotic stress within the developmental process (*Dunglas et al., 1999*).

In this study, erastin was used in an ex vivo culture model of tooth germ to investigate the possible role of ferroptosis during tooth morphogenesis. To definitively identify ferroptosis, multiple markers are needed, including lipid peroxidation, iron accumulation, mitochondria injury, and the expression of ferroptosis-related genes (*Stockwell, 2022*). According to our results, glutathione peroxidase 4 (Gpx4, a peroxidase to counter the oxidation of lipids in membranes) (*Yang et al., 2014*) is upregulated according to odontogenesis and amelogenesis, together with obvious accumulation of iron. Erastin significantly suppressed the morphogenesis of cultured tooth germ and showed a dose-dependent manner. Meanwhile, erastin-treated tooth germ has elevated lipid peroxidation indicated by expression of 4-hydroxy-2-nonenal (4-HNE, end-product of lipid peroxidation), enhanced iron accumulation, shrunken mitochondria, and alternated expression of ferroptosis related genes. However, all these phenomena could be partially rescued by ferrostatin-1 (Fer-1), a classical inhibitor of ferroptosis.

In summary, our results suggest that erastin induces morphological defects of tooth germ conducted by the activation of ferroptosis. Ferroptotic stress and its regulatory mechanism participate in tooth morphogenesis, which borders our understanding of the physiological function of non-apoptotic RCD in organ development.

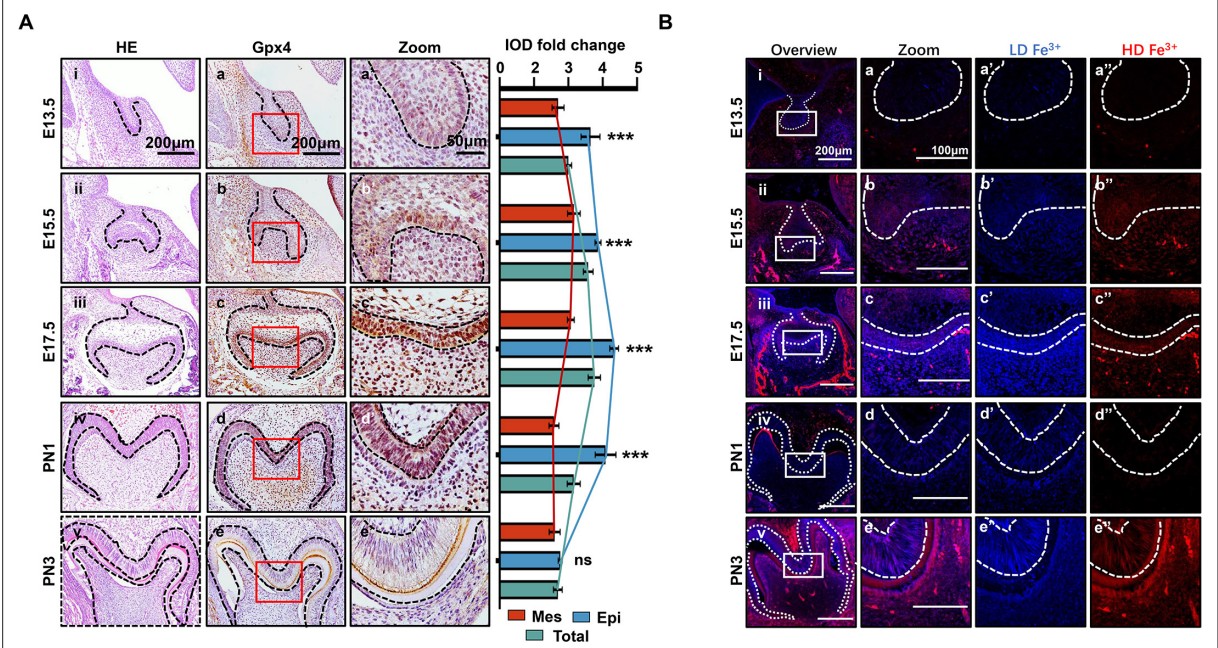

**Figure 1.** Spatiotemporal characterization of glutathione peroxidase 4 (Gpx4) expression and iron accumulation in tooth morphogenesis. (**A**) (**i–v**) HE staining for tooth germ in E13.5 to PN3, scale bars = 200 μm. (**a–e**) Gpx4 expression detected by immunohistochemistry (IHC), scale bars = 200 μm. (**a'–e'**) Enlarged view of each Gpx4 staining, scale bars = 50 μm; epithelia versus mesenchyme, n=3, ***p<0.001. (**B**) (**i–iv**) Iron probe staining in for tooth germ in E13.5 to PN3, scale bars = 200 μm. (**a–e**) Enlarged view of selected region, scale bars = 100 μm, low concentration (**a'–e'**, blue) and high concentration of iron (**a"–e"**, red) are present, scale bars = 100 μm. Epi, Epithelia; Mes, = mesenchyme.

The online version of this article includes the following source data and figure supplement(s) for figure 1:

**Source data 1.** Source data for the integrated optical density of glutathione peroxidase 4 (Gpx4) in epithelia, mesenchyme, and total area of tooth in *Figure 1A*.

**Figure supplement 1.** Negative control of IHC staining and the general view of ex vivo cultured tooth germ from D0 to D7.

## Results

### Spatiotemporal characterization of Gpx4 expression and iron accumulation in tooth morphogenesis

In the different developmental stages of the mouse's first mandibular molar, the expression of Gpx4 was detected by immunohistochemistry (IHC) (*Figure 1A*). These data revealed gradually upregulated expression of Gpx4 in tooth germ from embryonic day (E) 13.5 to E17.5 (*Figure 1Aa–c*), but started to decrease from PN1 to PN3 (*Figure 1Ad and e*). An enlarged view of each group indicated a different expression level of Gpx4 between dental epithelia and mesenchyme (*Figure 1Aa'–e'*). After calculating the integrated optical density (IOD) *Figure 1—source data 1* of Gpx4 in epithelia, mesenchyme, and total area of tooth germ, respectively, we found that Gpx4 in dental epithelia is higher than that in mesenchyme, although they both increased from E13.5 to E17.5, which could be observed in *Figure 1Aa'–d'*. Negative controls are listed in *Figure 1—figure supplement 1A*.

Iron accumulation, one of the core risk factors for ferroptosis, was detected by TPE-o-Py (*ortho*-substituted pyridinyl-functionalized tetraphenylethylene) (*Feng et al., 2018* using IF *Figure 1B*). The results showed accumulation of iron is upregulated within dental epithelium and mesenchymal cells during differentiation of odontoblast and ameloblast (*Figure 1Ba–c*) then decreased at PN1 which is consistent to the expression pattern of Gpx4. However, at PN3, the active secretory period of both ameloblast and odontoblast, the accumulation of iron rebound to a high level and distributed within enamel (*Figure 1Be*).

Collectively, our results demonstrated that at early stage of tooth development, enhancing risk factors of ferroptosis (iron accumulation) was accompanied by strengthening anti-ferroptosis mechanism (upregulation of Gpx4), and the accumulation of iron and the expression of Gpx4 in dental

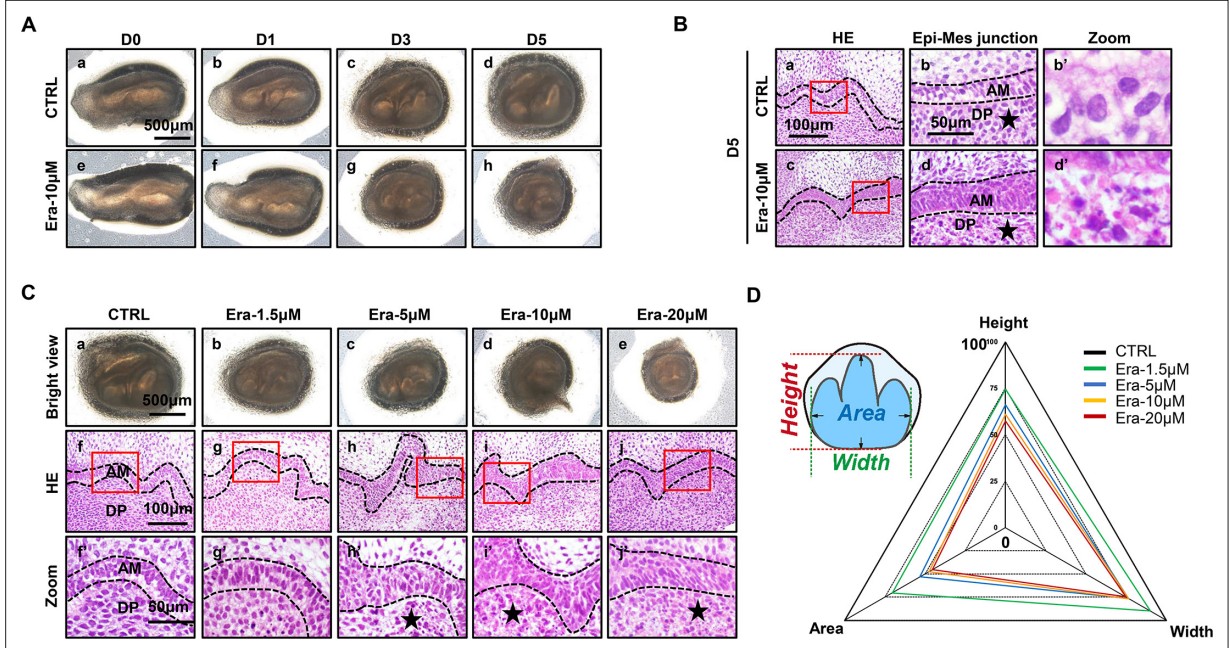

**Figure 2.** Erastin impairs tooth morphogenesis, especially within dental mesenchyme. (**A**) Gross anatomy of tooth germ cultured ex vivo for 5 d, scale bars = 500 μm. (**B**) (**a, c**) HE staining for tooth germ on day 5 (D5), scale bars = 100 μm. (**b, d**) High resolution of epi-mes junction papilla, scale bars = 50 μm; black stars point out necrotic-like cells (NLCs). (**b', d'**) NLCs indicated by black stars. (**C**) (**a–e**) Gross anatomy of tooth germs in different concentrations of erastin on D5, scale bars = 500 μm. (**f–j**) HE staining of different concentration-treated tooth germ on D5. (**f'–j'**) High-resolution view of epi-mes junction region of each tooth germ, scale bars = 50 μm. (**D**) Rada graph for calculation of height, width, and area in each tooth germ. Black dotted line outlines ameloblasts. AM, ameloblast; DP, dental papilla.

The online version of this article includes the following source data and figure supplement(s) for figure 2:

**Figure supplement 1.** Original bar graphs of height, width, and area of erastin-treated tooth germ.

**Figure supplement 1—source data 1.** Source data for the bar graphs in *Figure 2—figure supplement 1*.

epithelium/mesenchymal cells and extracellular matrix are critically related to the developmental stage.

## Erastin impairs tooth morphogenesis, especially within dental mesenchyme

To investigate the possible function of ferroptosis in tooth development, an ex vivo culture model of the molar germ was established as described before (*Jiang et al., 2017*), with or without treatment of erastin. Successful ex vivo culturing of tooth germ from D0 to D7 is presented in *Figure 1—figure supplement 1B*. All the molar germs were dissected from E15.5 mousses' mandibles and cultured in medium with or without erastin (10 μM) for 1, 3, and 5 d (*Figure 2A*). Comparing *Figure 2Ad and h*, gross anatomy showed apparent tiny tooth germ in the erastin-treated group than that of CTRL (Control group) on day 5 (D5). Histological analysis of D5 was performed in *Figure 2B*; treatment of erastin (10 μM) elevated the number of necrotic-like cells (NLCs), especially in dental mesenchyme (*Figure 2Bb' and d'*). We also found a dose-dependent manner of erastin in suppressing tooth morphogenesis (*Figure 2C*). In D5 incubation of different concentrations (1.5, 5, 10, and 20 μM) of erastin, tooth germ deteriorated morphologically (decreasing size, *Figure 2Ca–e*) and histologically (elevating number of NLCs within dental mesenchyme, *Figure 2Cf'–j'*) according to the increasing concentration of erastin of erastin treatment. Relative development suppression of tooth germs was then calculated by height, width, and coronal area compared to the CTRL (*Figure 2D*); the radar graph showed significant dose-dependent impairment of erastin to tooth germ, and the original bar graphs are listed in *Figure 2—figure supplement 1* (*Figure 2—figure supplement 1—source data 1*).

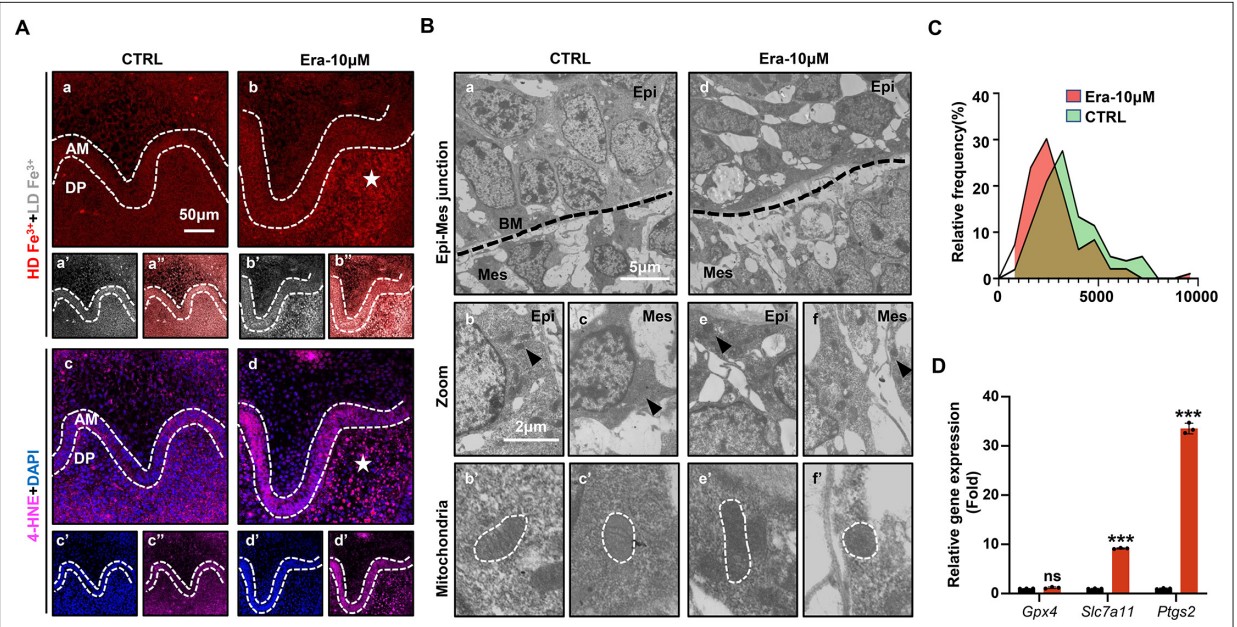

**Figure 3.** Ferroptosis is activated in erastin-treated molar germ. (**A**) (**a, b**) High-density $Fe^{3+}$ (red) in CTRL and Era-10 µM of day 5 (D5); white star points out strong fluorescence signal of $Fe^{3+}$, scale bars = 50 µm. (**a', b'**) Low-density $Fe^{3+}$ (gray) and (**a", b"**) merged view of iron probe staining. (**c, d**) Merged view of immunofluorescence (IF) staining of 4-hydroxy-2-nonenal (4-HNE) (magenta) and DAPI (blue), white star points out strong fluorescence signal of 4-HNE, scale bars = 5050 µm. (**c', d'**) For DAPI and (**c", d"**) 4-HNE. AM, ameloblast; DP, dental papilla; HD, high density; LD, low density. (**B**) Transmission electron microscope (TEM) scanning for CTRL and Era-10 µM on D5. (**a, d**) epi-mes junction area of CTRL and Era-10 µM on D5 are detected, scale bars = 5 µm. (**b, c**) Representative view of cells in epithelia and mesenchyme for CTRL, scale bars = 2 µm; black arrow points out typical mitochondria in each region (**b'**) for epithelia and (**c'**) mesenchyme, outlined by the white dotted line. (**e, f**) Representative view of cells in epithelia and mesenchyme for Era-10 µM, scale bars = 2 µm; black arrow points out typical mitochondria in each region (**e'**) for epithelia and (**f'**) mesenchyme, outlined by the white dotted line. (**C**) Relative frequency of the mitochondrial size in both groups. (**D**) Fold changes of gene expression in CTRL and Era-10 µM on D5 versus CTRL ***$p<0.001$.

The online version of this article includes the following source data and figure supplement(s) for figure 3:

**Source data 1.** Source data for the calculation of the mitochondrial area in *Figure 3C*.

**Source data 2.** Source data for the gene expression of *Gpx4, Slc7a11,* and *Ptgs2* in *Figure 3D*.

**Figure supplement 1.** Results for iron accumulation and 4-hydroxy-2-nonenal (4-HNE) expression in Era-10 µM of D1 and D3.

**Figure supplement 2.** Results for iron accumulation and 4-hydroxy-2-nonenal (4-HNE) expression in Era-1.5 µM, Era-5 µM, Era-10 µM, and Era-20 µM of D5.

## Ferroptosis is activated in erastin-treated molar germ

Accurate identification of ferroptosis should be determined by series markers, including iron concentration, lipid peroxidation, mitochondria dysmorphia, and overexpression of ferroptotic genes. We estimated the activation of ferroptosis in tooth germ. Indicated by TPE-o-Py detection in Era-10 µM of D5 (*Figure 3Aa–b"*), more iron is accumulated than that in CTRL and the region of high concentration mainly located in dental mesenchyme (red, *Figure 3Aa and b*). Lipid peroxidation was indicated by the expression of 4-HNE; the distribution pattern of 4-HNE was similar to iron accumulation, which suggested the activation of iron-dependent lipid peroxidation occurred within Era-10 µM of D5 (*Figure 3Ac–d"*). Results of D1 and D3 showed similar patterns in the accumulation of iron and the expression of 4-HNE (*Figure 3—figure supplement 1*), also in a dose-dependent manner on D5 (*Figure 3—figure supplement 2*). Morphological changes of mitochondria revealed severe shrinkage of mitochondria in dental mesenchyme of Era-10 µM (*Figure 3B*). The size of mitochondria in each group is calculated in *Figure 3C* (*Figure 3—source data 1*); results clearly showed the main size distribution of Era-10 µM is significantly decreased. Then, the expression of *Gpx4, Slc7a11,* and *Ptgs2* in each tooth germ was modulated by q-PCR. *Figure 3D* (*Figure 3—source data 2*) shows that *Ptgs2*, a gene representing the peroxidation level, dramatically increased in tooth germ after erastin treatment, while *Gpx4* and *Slc7a11*, anti-ferroptosis related genes, underwent upregulated but much more

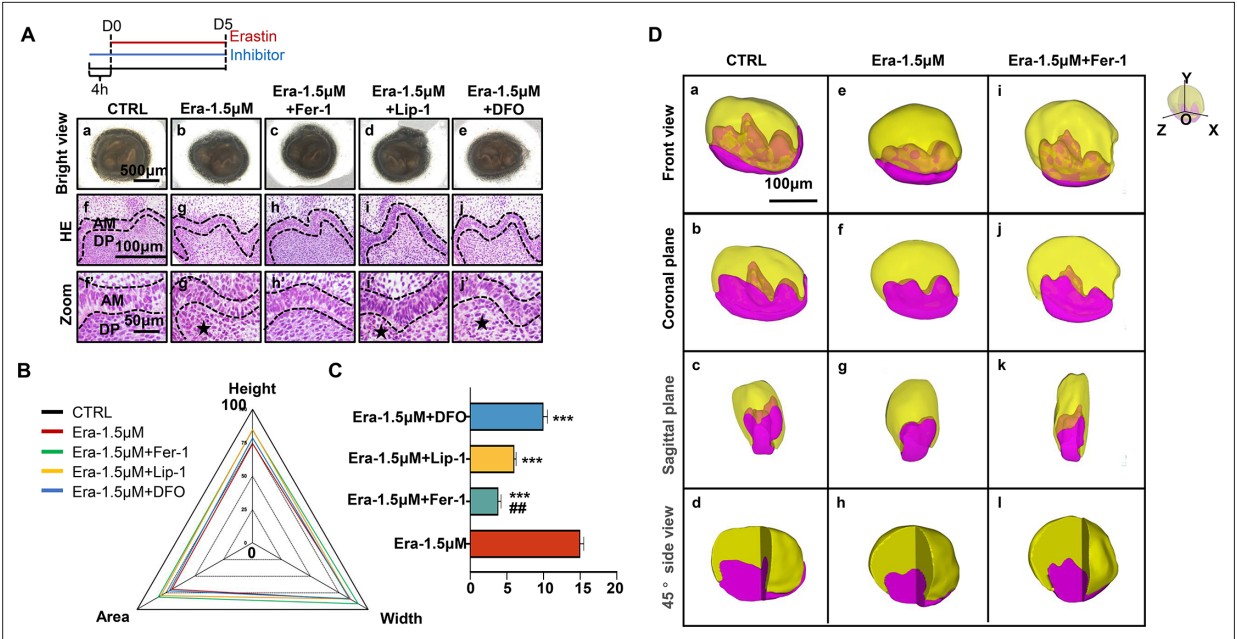

**Figure 4.** Ferroptotic inhibitors partially rescue erastin-impaired tooth organogenesis. (**A**) (**a–e**) Gross anatomy of tooth germs in the differently treated group, scale bars = 500 μm. (**f–j**) HE staining of differently treated tooth germ, scale bars = 100 μm. (**f'–j'**) High-resolution view of epi-mes junction region of each tooth germ; black dotted line outlines ameloblasts, black star points out necrotic-like cells (NLCs). AM, ameloblast; DP, dental papilla; scale bars = 50 μm. (**B**) Rada graph for calculation of height, width, and area in each tooth germ. (**C**) Average number of NLCs in each group versus Era-1.5 μM, \*\*\*p<0.001, versus Era-1.5 μM + Lip-1, ##p<0.01. (**D**) 3D reconstructed view of tooth germ on day 5 (D5). (**a–d**) CTRL from the front view, coronal plane, sagittal plane, and 45° side view. (**e–h**) Era-1.5 μM and (**i–l**) Era-1.5 μM + Fer-1 are viewed the same way. Scale bars = 100 μm.

The online version of this article includes the following source data and figure supplement(s) for figure 4:

**Source data 1.** Source data for the bar graphs in *Figure 4C*.

**Figure supplement 1.** Original bar graphs of height, width, and area of tooth germ in rescue assay.

**Figure supplement 1—source data 1.** Source data for the bar graphs in *Figure 4—figure supplement 1*.

**Figure supplement 2.** Source sequential HE slides for 3D reconstruction of CTRL.

**Figure supplement 3.** Source sequential HE slides for 3D reconstruction of Era-1.5 μM.

**Figure supplement 4.** Source sequential HE slides for 3D reconstruction of Era-1.5 μM + Fer.

moderate expression. All these results contributed to the confirmation of the activated ferroptosis occurred in erastin-treated molar germ and demonstrated that no lethal concentration of erastin could lead to tooth morphogenesis partially through activation of ferroptosis in dental mesenchyme.

## Ferroptotic inhibitors partially rescue erastin-impaired tooth organogenesis

To further prove the function of ferroptosis in the suppression of tooth germ morphogenesis, a rescue assay was applied using Fer-1 (classical inhibitor of ferroptosis), liproxstatin-1 (Lip-1) (inhibitor of lipid peroxidation), and deferasirox (DFO) (iron chelator) to co-incubated tooth germ with erastin.

As the results show in *Figure 4A*, although all these three molecules show rescuing efficiency, Fer-1 holds the highest efficiency in recovering the organogenesis of molar germ in gross anatomy (*Figure 4Ac*) and reducing the number of NLCs at the histological level (*Figure 4Ah and h'*). Calculating the height, width, area of tooth germ (*Figure 4B*; original bar graphs are listed in *Figure 4—figure supplement 1* and *Figure 4—figure supplement 1—source data 1*), and the number of NLCs (*Figure 4C*, *Figure 4—source data 1*) showed inhibitors could partially rescue impairment by erastin, while Fer-1 rescued most.

To avoid bias caused by sectioning, sequential HE slides of CTRL and Era-1.5 μM were applied for 3D reconstruction as previously described (*Wu et al., 2020*). In *Figure 5Da–j*, tooth germ in CTRL showed well-developed morphology in the 3D models. However, cusp formation in Era-1.5

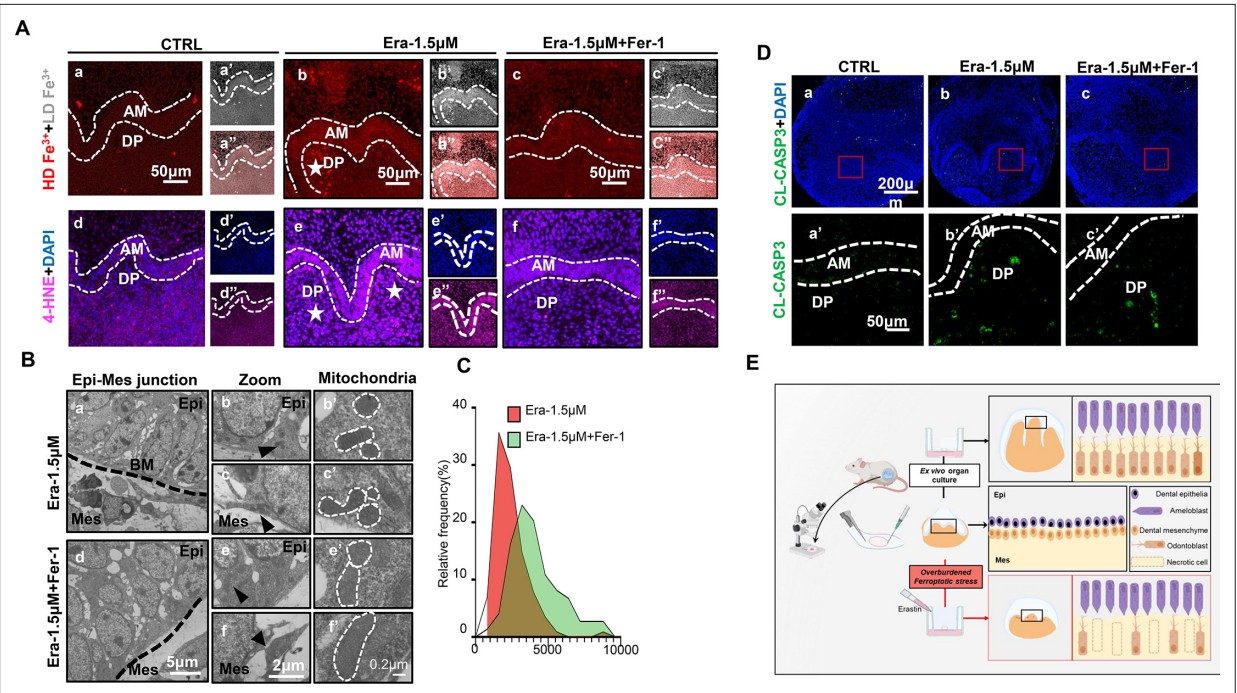

**Figure 5.** Ferroptosis is the dominant cell death type contributing to erastin-impaired tooth morphogenesis. (**A**) (**a–c**) High-density $Fe^{3+}$ (red) in CTRL, Era-1.5 µM, and Era-1.5 µM + Fer-1 of day 5 (D5), white star points out strong fluorescence signal of $Fe^{3+}$, scale bars = 50 µm. (**a'–c'**) Low-density $Fe^{3+}$ (gray) and (**a''–c''**) merged view of iron probe staining. (**d–f**) Merged view of immunofluorescence (IF) staining of 4-hydroxy-2-nonenal (4-HNE) (magenta) and DAPI (blue), white star points out strong fluorescence signal of 4-HNE, scale bars = 50 µm. (**d'–f'**) DAPI and (**d''–f''**) 4-HNE, AM, ameloblast; DP, dental papilla; HD, high density; LD, low density. (**B**) Transmission electron microscope (TEM) scanning of Era-1.5 µM and Era-1.5 µM + Fer-1 on D5. (**a, d**) epi-mes junction area of Era-1.5 µM and Era-1.5 µM + Fer-1 on D5 is detected, scale bars = 5 µm. (**b, c**) Representative view of cells in epithelia and mesenchyme for Era-1.5 µM, scale bars = 2 µm; black arrow points out typical mitochondria in each region (**b'**) for epithelia and (**c'**) mesenchyme, outlined by the white dotted line. (**e, f**) Representative view of cells in epithelia and mesenchyme for Era-1.5 µM + Fer-1, scale bars = 2 µm; black arrow points out typical mitochondria in each region (**e'**) for epithelia and (**f'**) mesenchyme, outlined by the white dotted line. (**C**) Relative frequency of the mitochondrial size in both groups. (**D**) (**a–c**) Expression of CL-CASP3 (green) in CTRL, Era-1.5 µM, and Era-1.5 µM + Fer-1 on D5, scale bars = 200 µm. (**a'–c'**) Enlarged view of CL-CASP3 in each group, scale bars = 50 µm. (**E**) Schematic model illustrates the overburdened ferroptotic stress impaired tooth morphogenesis in ex vivo organ culture model.

The online version of this article includes the following source data and figure supplement(s) for figure 5:

**Source data 1.** Source data for the calculation of the mitochondrial area in *Figure 5C*.

**Figure supplement 1.** Activation of apoptosis indicated by CL-CASP3 and TUNEL staining.

**Figure supplement 1—source data 1.** Source data for the bar graphs in *Figure 5—figure supplement 1*.

µM (*Figure 4De–h*) underwent significant suppression compared with CTRL in different directions of view; original sequential slides are listed in *Figure 4—figure supplement 2* for CTRL and *Figure 4—figure supplement 3* for Era-1.5 µM. 3D reconstruction was also performed in Era-1.5 µM + Fer-1 (*Figure 4Di–l*). Compared to Era-1.5 µM (*Figure 4De–h*), Fer-1 reversed abnormal features dramatically in tooth morphology and cusp formation. Source sequential sections are listed in *Figure 4—figure supplement 4*.

## Ferroptosis is the dominant cell death type contributing to erastin-impaired tooth morphogenesis

We further estimated the key characteristics of ferroptosis to assess whether Fer-1 rescued erastin-impaired tooth germ by inhibiting ferroptosis. Comparing with the CTRL group after 5 d ex vivo culture (*Figure 5Aa and d*), iron accumulation and lipid peroxidation (indicated by 4-HNE) in Era-1.5 µM is much higher (*Figure 5Ab and e*). Fer-1 treatment vanished iron accumulation and lipid peroxidation caused by erastin (*Figure 5Ac and f*). Results of transmission electron microscope (TEM) clearly showed decreased mitochondria shrinkage in Era-1.5 µM + Fer-1 than that of Era-1.5 µM (*Figure 5B*

*and C*, *Figure 5—source data 1*). All these data convinced that Fer-1 rescued erastin-impaired tooth morphogenesis by inhibiting ferroptosis.

Apoptosis is the main innate cell death type in physiological process. To exclude possible overactivation of apoptosis induced by erastin treatment, we detected CL-CASP3 by IF staining, main executor protein of apoptosis, in each group. In *Figure 5D*, CL-CASP3 is weakly expressed in the CTRL group since the activation of apoptosis is physiologically required in tooth development (*Figure 5D*). The expression of CL-CASP3 slightly increased in both Era-1.5 µM and Era-1.5 µM + Fer-1, but showed no statistical differences compared with the CTRL group (*Figure 5—figure supplement 1A*, *Figure 5—figure supplement 1—source data 1*). TUNEL assay is also performed to identify apoptotic cells by detecting DNA damage, and results further convinced that apoptosis is not significantly activated in erastin-treated tooth germ (*Figure 5—figure supplement 1B*). Taken together, ferroptosis is the dominant cell death led by erastin treatment in impaired tooth germ.

## Discussion

In the last several decades, the beneficial role of apoptosis in regulating organ development and tissue regeneration has been identified (*Singh et al., 2019*). Apoptosis in tooth development had been characterized by the activation of Caspase 3 and DNA damage, which revealed a spatiotemporal apoptotic cell death pattern due to different stages of tooth morphogenesis (*Shigemura et al., 2001*). This raises the question of whether other newly determined non-apoptotic cell death pathways also participate in the physiological processes like development, maintaining homeostasis, aging, etc. Except for NETosis, a neutrophil extracellular trap-related cell death, the possible function and mechanism of the rest of the non-apoptotic cell death including pyroptosis, necroptosis, and ferroptosis are still barely investigated. Characterized by its close relationship with the metabolism of lipid, iron, and ROS (*Bonadonna et al., 2022*; *Bowers et al., 2020*), risk factors inducing ferroptotic stress and/or activating ferroptosis are also critically involved in development and aging, which made ferroptosis and its regulatory mechanism, other than apoptosis, pyroptosis, and necroptosis, a promising undiscovered type of cell death during tooth development.

To explore the potential involvement of ferroptosis in organogenesis, we have developed an ex vivo culture model of tooth germ morphogenesis. This system allows for the application of erastin to induce significant activation of ferroptosis throughout the developmental process and offers a valuable opportunity to investigate the specific mechanisms underlying the role of ferroptosis in organ development. Our study detected the expression of Gpx4 and the accumulation of iron both in mouse mandibular incisor and developing first molar. Although Gpx4 is expressed ubiquitously in both tooth germ and incisor of mouse, its abundance differs in different cell types. Results revealed an increasing iron accumulation both in odontoblast and ameloblast according to the developmental process. The spatiotemporal expression of Gpx4 is also positively linked to odontogenesis and amelogenesis. This phenomenon indicated growing ferroptotic stress (accumulating iron) and strengthening anti-ferroptosis mechanism (increasing expression of Gpx4) physiologically coexisted and may maintain a critical balance along with the differentiation and maturation of odontoblast and ameloblast. However, the discovery of changes in iron and lipid metabolism during tooth morphogenesis is not novel. In the 1930s, pioneer scientists in dental biology had already identified the presence of iron in the tooth of different animals (*Rosebury, 1934*; *Ratner, 1935*; *Suga et al., 1992*), and then found some defects of enamel in mouse are related to abnormal iron metabolism (*Puri et al., 2015*). Lipid metabolism and lipid peroxidation, the other core risk factors of ferroptosis, were also described in the early stage of dental biology research (*Dunglas et al., 1999*; *Goldberg et al., 1995*; *Yoshida et al., 2012*). Even when the risk factors of ferroptosis had been reported to participate in tooth development, there are still no reports about the exact ferroptosis-related tooth developmental defects. Our results provided a new perspective to reconsider the underlying function of all these ancient studies in a comprehensive manner. They illustrated the importance of the Gpx4-dependent anti-ferroptosis pathway in managing all these already existing ferroptotic stress during tooth morphogenesis. Future in vivo studies utilizing transgenic mice are needed to systematically analyze the role of ferroptosis/ferroptotic stress during tooth and other organ development.

To further investigate the meaning of this precarious balance between ferroptotic stress and expression of Gpx4, we use erastin to inhibit the internalization of GSH, which is the critical substrate for Gpx4 to protect cells from lipid peroxidation. The developmental role of Gpx4 had been studied even

before the ferroptosis was formally described (before 2012). In situ hybridization indicated expression of Gpx4 in all developing germ layers during gastrulation and in the somite stage in the developing central nervous system and in the heart (*Borchert et al., 2006*), which made Gpx4 (-/-) mice die embryonically in utero by midgestation (E7.5) and are associated with a lack of normal structural compartmentalization (*Yant et al., 2003*). Specific deletion of Gpx4 during developmental process was found to participate in the maturation and survival of cerebral and photoreceptor cell (*Wirth et al., 2010*; *Ueta et al., 2012*). In recent years, more ferroptosis-related functions of Gpx4 were discovered in neutrophil (*Li et al., 2021*) and chondrocyte (*Wang et al., 2022*) of adult mice, in which specific deletion will lead to ferroptosis-induced organ dysregulation and degeneration. Thus, it is essential to assess the biological function Gpx4-dependent ferroptotic suppressing system. In our study, erastin significantly impaired tooth morphogenesis in a dose-dependent manner within the ex vivo tooth germ culture model. The ex vivo organ culture of tooth germ is a well-established classical model for the study of tooth morphogenesis. Compared to in vivo study, the ex vivo culture of tooth germ is convenient for manipulating culture conditions and investigating factors affecting tooth germ in a high-throughput way (*Nakao et al., 2007*). Although lacking circulation and immune system, the ex vivo culture of the tooth germ, other than the traditional 2D culture of dental cells in vitro, can retain most properties of tooth development, like interactions among oral epithelia, mesenchyme, and stromal cells. We successfully established this model, and the tooth germs from D0 to D7 are well developed (*Figure 1—figure supplement 1B*). To induce ferroptosis, erastin is the most used agent inhibiting GSH transport but is not stable in vivo, which makes ex vivo organ culture of tooth germ the ideal way to study the possible function of ferroptosis/ferroptotic stress in tooth morphogenesis. Moreover, according to our results, erastin treatment will not induce overactivation of apoptosis in tooth germ (*Figure 5D*).

The histological analysis by HE staining showed an increased number of NLCs located in dental mesenchyme of erastin-treated tooth germ. 3D reconstruction of all the slides convinced that necrotic mainly occurred in the region of the odontoblast layer. Unlike CL-CASP3 to apoptosis, membrane localization of the GSDM family proteins and MLKL to pyroptosis and necroptosis, respectively, ferroptosis is a special type of cell death which has unique inducer but no special proteins reflecting its activation. Thus, ferroptosis in erastin-treated tooth germ is determined by the accumulation of iron, upregulation of 4-HNE, shrunken and condensed mitochondria, dramatically upregulated expression of *Psg2* (risk marker to ferroptosis), and mildly increased expression of *Gpx4, Slc7a11* (protective factor to ferroptosis). These results indicated that erastin could lead to abnormal tooth morphology via activating ferroptosis. Moreover, characterized by more apparent NLCs, stronger iron and 4-HNE fluorescence signal, and severer mitochondria degeneration, dental mesenchyme cell/odontoblast seems more sensitive to erastin-induced ferroptosis than that of dental epithelium cell/ameloblast.

According to our results, ferroptotic stress is physiologically increased during tooth development. Thus, rather than 'initially activate ferroptosis' in tooth morphogenesis, erastin, a system $x_c^-$ inhibitor, inhibits GSH production and induced ferroptosis in tooth germ is more likely led by 'overburdened ferroptotic stress.' As a multistep process of cell death, the different inhibitor has a different target to suppress ferroptosis. In this study, regular ferroptosis inhibitors were applied in the rescue assay of impaired tooth morphogenesis. Tooth germ was treated by Fer-1, Lip-1 (radical trapping agents that inhibit the propagation of lipid peroxidation), and DFO (an iron chelator). Surprisingly, although all these inhibitors could reverse erastin-induced impairment of tooth morphogenesis, Fer-1 reduced the number of NLCs much more efficiently than lip-1 and DFO. Different from lip-1 or DFO, which only targets lipid peroxidation or labile iron accumulation, Fer-1 could both inhibit lipid peroxidation and reduce the labile iron pool in cells, and is notably not consumed while inhibiting iron-dependent lipid peroxidation (*Miotto et al., 2020*). Sustaining double effects of Fer-1 in inhibiting ferroptosis possibly enables it to effectively rescue the impairment of tooth germ than other agents.

In conclusion, using ex vivo culture model of tooth germ, we identified a continuing accumulation of ferroptotic stress in both odontogenesis and amelogenesis during tooth development. Activation of ferroptosis impaired tooth morphogenesis, especially within dental mesenchyme, and could be partially rescued by ferroptotic inhibitor. This study provides a promising model to effectively investigate the developmental role of ferroptosis and will broaden our knowledge about the possible involvement of non-apoptotic RCD during organogenesis.

# Materials and methods

## Animals and organ culture

ICR mice were purchased from Chengdu Dossy Experimental Animals Co., Ltd (Sichuan, China). All animal work was done according to the National Institutes of Health guidebook and approved by the Committee of the Ethics of Animal Experiments of Sichuan University (WCHSIRB-D-2021-12544).

The presence of a vaginal plug was used as an indication of embryonic day 0 (E0). The first mandibular molar tooth germs of E15.5 were dissected under stereomicroscope (Carl Zeiss, Germany); each experimental group contains at least three tooth germs from different mouse embryos. All the dissection steps were performed on ice. The dissected tooth germs were randomly placed in the upper chamber of the Trans-well (3450, Corning, USA) and four tooth germs were placed on one dish.

The culture medium and drugs were mixed and placed in the lower chamber. Tooth germs were cultured in DMEM/F12 supplemented with 10% fetal bovine serum, 50 U/ml penicillin/streptomycin, and 100 μg/ml ascorbic acid and incubated at 37°C and 5% $CO_2$. Medium was changed every other day. Tooth germs were cultured for 5 d in the presence of different concentration of erastin (0 μM, 1.5 μM, 5 μM, 10 μM, and 20 μM). Culture medium added with Fer-1 (1 μM), Lip-1 (200 nM), and DFO (100 μM) was used to pretreat molar germ for 4 hr. All these small molecules were purchased from Selleck. After 5 d of cultivation, the tooth germs were fixed in 4% PFA and embedded in paraffin. Tooth germs were photographed at 0 d, 1 d, 3 d, and 5 d, and each tooth size (width, height, and area) were measured (n = 9 for each group) using the modified method reported in the literature previously. Each experiment was repeated three times (*Jiang et al., 2019*).

## Immunohistochemistry

Sections were dewaxed and rehydrated before antigen repair, and then cooled to room temperature. Sections were incubated overnight at 4°C with Gpx4 (1:200, Invitrogen, PA5-102521) in 2% bovine serum albumin (BSA)/phosphate-buffered saline (PBS), pH 7.4. Negative control sections were incubated with 2% BSA/PBS. After washing, sections were then incubated with SignalStain Boost IHC Detection Reagent (8114S, Cell Signaling Technology, USA), then washed and incubated with diaminobenzidine (DAB) to detect any reactions, and then examined by light microscopy after counterstaining with hematoxylin.

## Immunofluorescence and iron probe staining

Tissue sections were treated with PBS and 0.5% Triton X-100 for permeabilization. After washing and blocking, sections were incubated with 4-HNE (1:200, Abcam, ab48506) and cleaved Caspase 3 (1:200, Cell Signaling Technology, 9664) in blocking buffer (PBS and 2% BSA) overnight at 4°C. After washing, second antibody Alexa Fluor 488 (1:200, Abcam, ab150077) and Alexa Fluor 594 (1:200, Abcam, ab150116) mixed with 4′,6-diamidino-2-phenylindole (DAPI) was applied. Then they were incubated at room temperature for 1–2 hr and washed, and the slices were sealed.

To accurately detect the changes in iron accumulation within differently cultured tooth germs, we used an aggregation-induced emission featured iron (III) probe of TPE-o-Py (*Feng et al., 2018*), which was kindly gifted by Prof. Youhong Tang. This probe displays high sensitivity and selectivity toward iron (III) detection. The recognition arises from the position isomer of ortho-substitution and the fact that TPE-o-Py has a low acid dissociation constant (pKa) that is close to that of hydrolyzed $Fe^{3+}$. The iron probe staining was performed as described. Briefly, TPE-o-Py was dissolved in tetrahydrofuran before being added to PBS and diluted to working concentration of 20 μM. The working solution was placed in sections and incubated at room temperature for half an hour and aspirated and sealed, then a laser confocal microscope (Olympus FV3000, Japan) was used to detect the fluorescence signal. Since the TPE-o-Py probe pronounced red shift in fluorescence emission that is positively related to the concentration of iron, low concentration of $Fe^{3+}$ was detected under fluorescence channel of Alexa Fluor 405 (excitation wavelength: 402 nm; emission wavelength: 421 nm), while Alexa Fluor 594 (excitation wavelength: 590 nm; emission wavelength: 617 nm) for high concentration.

## Tissue preparation for transmission electron microscope

Prefixed with a 3% glutaraldehyde, the tissue was then postfixed in 1% osmium tetroxide, dehydrated in series acetone, infiltrated in Epox 812 for longer, and embedded. The semithin sections were stained with methylene blue, and ultrathin sections were cut with a diamond knife and stained

with uranyl acetate and lead citrate. Sections were examined with JEM-1400-FLASH Transmission Electron Microscope.

## RNA extraction and qPCR

Total RNA of tooth germs (E15.5 cultured for 5 d) were extracted using TRIzol Reagent (Invitrogen, USA) according to manufacturer's instructions. The amount and integrity of RNA were assessed by measurement of absorbance at 260 and 280 nm. First-strand cDNA synthesis was performed with a HiScript III Q RT SuperMix for qPCR (Vazyme, China). The levels of *Gpx4, Ptgs2,* and *Slc7a11* were measured by quantitative real-time PCR (Bio-Rad, USA) with ChamQ Universal SYBR qPCR Master Mix (Vazyme, China) and normalized to the level of β-actin mRNA. These experiments were performed in triplicate. The primer sequences used in qPCR are listed below.

| Gene | Sequence |
| --- | --- |
| | F: CCTCCCCAGTACTGCAACAG |
| *Gpx4* | R: GGCTGAGAATTCGTGCATGG |
| | F: CTGCGCCTTTTCAAGGATGG |
| *Ptgs2* | R: GGGGATACACCTCTCCACCA |
| | F: GATGGTCCTAAATAGCACGAGTG |
| *Slc7a11* | R: GGGCAACCCCATTAGACTTGT |
| | F: AGATGTGGATCAGCAAGCAG |
| *Actb* | R: GCGCAAGTTAGGTTTTGTCA |

## 3D reconstruction of tooth germ sections

Sequential section and subsequent HE staining of each tooth germ were performed as described before. Digital pathological system (Olympus vs200) was used to scan all the stained sections and reconstructed by following the previously described protocol (*Wu et al., 2020*).

## TUNEL assay

We used a One Step TUNEL Apoptosis Assay Kit (Beyotime, C1090) to detect the possible DNA damage in apoptotic cell of tooth germs. The TUNEL assay was performed following instructions.

## Statistical analysis

Analysis of Gpx4 relative expression levels was carried out with Image-pro plus7.0 (Media Cybernetics, USA). Statistical analyses were carried out using the GraphPad Prism version 8.00 (GraphPad Software, San Diego, CA). All statics are shown as the arithmetic mean ± standard error of the mean. The significance of differences between groups was tested using the one-sample *t*-test. Differences were considered significant when $p<0.05$. All the experiments were independently replicated three times.

## Acknowledgements

This study was funded by grants from the National Natural Science Foundation of China U21A20368 (LY), 82101000 (HW), and 82201045 (FY), and by the Young Elite Scientist Sponsorship Program by China Association for Science and Technology 2022QNRC001 (FY). The iron probe, TPE-*o*-Py, was kindly gifted by Prof. Youhong Tang (Flinders University, Australia) and Prof. Benzhong Tang (the Hong Kong University of Science and Technology, China).

## Additional information

### Funding

| Funder | Grant reference number | Author |
|---|---|---|
| National Natural Science Foundation of China | U21A20368 | Ling Ye |
| National Natural Science Foundation of China | 82101000 | Haisheng Wang |
| National Natural Science Foundation of China | 82201045 | Fanyuan Yu |
| China Association for Science and Technology | 2022QNRC001 | Fanyuan Yu |

The funders had no role in study design, data collection and interpretation, or the decision to submit the work for publication.

### Author contributions

Haisheng Wang, Conceptualization, Resources, Data curation, Software, Formal analysis, Funding acquisition, Validation, Investigation, Methodology, Writing - original draft, Writing - review and editing; Xiaofeng Wang, Conceptualization, Resources, Software, Formal analysis, Investigation, Methodology, Writing - original draft; Liuyan Huang, Resources, Data curation, Validation, Methodology, Writing - review and editing; Chenglin Wang, Writing - review and editing; Fanyuan Yu, Conceptualization, Resources, Data curation, Software, Supervision, Funding acquisition, Validation, Investigation, Methodology, Project administration, Writing - review and editing; Ling Ye, Conceptualization, Data curation, Supervision, Funding acquisition, Validation, Project administration, Writing - review and editing

### Author ORCIDs

Haisheng Wang (ID) http://orcid.org/0000-0002-5133-9587
Fanyuan Yu (ID) http://orcid.org/0000-0002-6879-3508

### Ethics

This study was performed in strict accordance with the recommendations in the Guide for the Care and Use of Laboratory Animals of the National Institutes of Health. All animal work was done according to the National Institutes of Health guidebook and approved by the Committee on the Ethics of Animal Experiments of Sichuan University (WCHSIRB-D-2021-12544). All surgery was performed under sodium pentobarbital anesthesia, and every effort was made to minimize suffering.

Reviewer #1 (Public Review): https://doi.org/10.7554/eLife.88745.3.sa1
Reviewer #2 (Public Review): https://doi.org/10.7554/eLife.88745.3.sa2
Reviewer #3 (Public Review): https://doi.org/10.7554/eLife.88745.3.sa3
Author Response https://doi.org/10.7554/eLife.88745.3.sa4

## Additional files

### Supplementary files
• MDAR checklist

### Data availability

All data generated or analysed during this study are included in the manuscript and supporting file; Source Data files have been provided for all the figures.

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
