## [Editor Report · eLife assessment]

This **important** study time elegantly demonstrates that ferroptotic stress may play critical roles in regulating tooth germ development. The evidence presented is **compelling**, based on an explant model and providing novel mechanistic insights into tooth development.

---

## [Referee Report · Reviewer #1 (Public Review)]

Cell death plays a critical role on regulating organogenesis. During tooth morphogenesis, apoptosis of embryonic dental tissue plays critical roles on regulating tooth germ development. The current study focused on ferroptosis, another way of cell death which has rarely been investigated in tooth development, and showed it may also play an important role on regulating the tooth dimension. The topic is novel and interesting, but the experimental design has some flaws which compromised the study.

The entire study was based on ex vivo tooth germ explant culture. I hope the authors can continue working on this direction with more convincing transgenic models.

---

## [Referee Report · Reviewer #2 (Public Review)]

The present study by Ye et al. characterizes some of the major effects of ferroptotic stress on tooth morphogenesis.

The strengths of this study are its innovative nature and the beautiful histology. Mechanistic data are convincing Overall, the study is well done.

---

## [Referee Report · Reviewer #3 (Public Review)]

This is an interesting work reporting ferroptosis that is involved in the tooth morphogenesis. The authors showed that Gpx4, the core anti-lipid peroxidation enzyme in ferroptosis, is upregulated in tooth development using ex vivo culture system.

---

## [Author Response]

The following is the authors’ response to the original reviews.

**Public Reviews:**

**Reviewer #1 (Public Review):**
Cell death plays a critical role on regulating organogenesis. During tooth morphogenesis, apoptosis of embryonic dental tissue plays critical roles on regulating tooth germ development. The current study focused on ferroptosis, another way of cell death which has rarely been investigated in tooth development, and showed it may also play an important role on regulating the tooth dimension. The topic is novel and interesting, but the experimental design has many flaws which significantly compromised the study.1. The entire study was based on ex vivo tooth germ explant culture. Mandibular tooth germs of E15.5 (bell stage) were isolated for ex vivo culture. Most tooth germ explant culture experiments were actually using tooth germ of much earlier stages (E11.5-E13.5) for organ culture. After E16.5, both the large size and initially formed enamel/dentin could prevent nutrition from penetrating inside. Also, using tooth germ of earlier stage will help identify impact of ferroptosis upon early tooth development.1. Due to limited penetration, the ex vivo culture in the study lasted for no more than 5 days. I would recommend the authors to perform kidney capsule transplantation as an alternative approach, which can support tooth germ development much longer even into root formation.1. The major justification of using tooth germ ex vivo culture as the model in the study was to "conduct high-throughput analysis". However, the study could hardly be qualified as a high-throughput analysis. I would recommend the authors perform RNA sequencing for comparing tooth germs before/after erastin treatment. Such experiments won't take too much time or resource.

We are grateful for the insightful feedback on our ex vivo tooth germ culture model. We initially chose the E15.5 tooth germ over earlier stages due to peak Gpx4 expression and iron accumulation during molar development, which occurs between E15.5 and E17.5 (Figure 1A & 1B). This period may be the most sensitive to ferroptotic stress during tooth development. Our experiments also demonstrated that the tooth germ displays robust growth after seven days of ex vivo cultivation (Figure supplement 1B).

Kidney capsule transplantation is indeed an ideal method for ex vivo tooth germ culture. However, in our studies, we used erastin – a classic ferroptosis inducer – which exhibits instability in vivo, thereby constraining our investigation using kidney capsule transplantation.

Our results about Gpx4 expression in the tooth germ during development (Figure 1A) showed a spatiotemporal pattern. This pattern suggests that bulk RNA sequencing of the tooth germ might not yield accurate revelations about changes in ferroptosis-related genes. We are presently using transgenic mice to further study the impact of excessive in vivo ferroptotic stress on tooth development. In these experiments, we intend to conduct single-cell RNA sequencing to explore detailed alterations in the tooth germ.

1. Although the study mostly used molars as the model, the in vivo iron concentration was only demonstrated on incisors, but not molars (Figure 1).

We have updated Figure 1B to include images of molars, which illustrate the accumulation of iron during molar development. The iron concentration peaks at E17.5, then decreases at PN0. Interestingly, unlike Gpx4 expression, iron accumulation rebounds at PN3. To gain a more accurate understanding, further in vivo studies utilizing transgenic mice are required.

1. Phenotype analysis in Figure 2 is too superficial. Only dimensional information was provided. Cusps number, cusps distribution pattern and rooth/furcation formation were not evaluated. Differentiation of ameloblast/odontoblast was not evaluated. The proliferation rate in the dental epithelium/mesenchyme was not analyzed.

The cusps number/distribution pattern are not influenced by erastin treatment in recent model (Figure 2A & 2C). Recent ex vivo culture model of tooth germ is unable to investigate the possible function of ferroptotic stress in rooth/furcation formation since it mainly initiates from PN4 to PN7. The proliferation and differentiation of dental epithelium/mesenchyme will be analyzed using transgenic mice in vivo.

1. Low magnification images should be included in Figure 3 to display the entire tooth germs.

The emission spectrum of recent utilized iron probe will extend due to increasing concentration of iron. This property makes the counter staining of tissue samples unavailable. The structure of the ex vivo cultured tooth germ could only be recognized in high magnification. The calculation could represent the entire alternation.

1. In Figure 4, does ferroptotic inhibitor eliminate the iron accumulation in the tooth germ? How about the expression level of several target genes shown in Figure 3?

In Fig 5, Fer-1 reduced the iron accumulation in tooth germ. Different inhibitors suppressed ferroptosis via different ways, Lip-1 mainly inhibits lipid peroxidation, DFO is an iron chelator which reduces the labile iron pool, Fer-1 is reported to both inhibit lipid peroxidation and reduce the labile iron pool, their functions to the accumulation of iron might be varied. The core risk factors of ferroptosis are lipid peroxidation and iron accumulation, thus in Fig 5, we analyzed the expression of 4HNE and the accumulation of iron to illustrated the suppression o ferroptosis instead of detecting several regulatory genes.

1. The manuscript has many typos and grammar mistakes. All "submandibular" should be simply "mandibular". "eastin" should be "erastin" (line 92). "partly" should be "partially" (line 611).

We addressed all the gramma and typo errors.

**Reviewer #2 (Recommendations for The Authors):**
This is a very well done study. However, writing is absolutely substandard. The authors should check and review extensively for improvements to the use of English. This is not just about language but also about style of the paper and presentation. As written, the abstract is not concise at all, and the overall logic of the study is not well presented. Currently, the abstract reads like another introduction.

We improved our presentation.

**Reviewer #3 (Recommendations for The Authors):**
This is an interesting work reporting ferroptosis that is involved in the tooth morphogenesis. The authors showed that Gpx4, the core anti-lipid peroxidation enzyme in ferroptosis, is upregulated in tooth development using ex vivo culture system. They convincingly demonstrated that ferroptosis, but apoptosis, was present in tooth morphogenesis. The findings are interesting and novel. The work represents one of the earliest works studying Ferroptosis in tooth morphogenesis. There are several minor concerns.1. The abstract is too long and should be shortened.

We modified the abstract to make it concise.

1. Can the Gpx4 quantitatively be measured by qRT-PCR?1. How is Gpx4 regulated during development? If unknown, the authors should discuss it at least1. Are there any tooth developmental defects associated with ferroptosis? If there is one, the authors should discuss it.

Our research on Gpx4 expression in the tooth germ during development (Figure 1A) highlights a specific spatiotemporal pattern. This pattern suggests that bulk RNA sequencing of the tooth germ may not provide accurate insight into changes in ferroptosis-related genes.

The developmental role of Gpx4 had been studied even before the ferroptosis was formally described (before 2012). In situ hybridization indicated expression of Gpx4 in all developing germ layers during gastrulation and in the somite stage in the developing central nervous system and in the heart, which made Gpx4 (-/-) mice die embryonically in utero by midgestation (E7.5) and are associated with a lack of normal structural compartmentalization. Specific deletion of Gpx4 during developmental process were found to participate in the maturation and survival of cerebral and photoreceptor cell. Recent years, more ferroptosis related function of Gpx4 were discovered in neutrophil and chondrocyte of adult mice, in which specific deletion will lead to ferroptosis-induced organ dysregulation and degeneration.

At present, no systematic study has been conducted on ferroptosis or ferroptotic stress in relation to tooth developmental defects. However, as early as the 1930s, pioneering dental biologists had already identified the presence of iron in the teeth of various animals. They also found that some enamel defects in mice were related to abnormal iron metabolism. Lipid metabolism and lipid peroxidation, which are other key risk factors of ferroptosis, were also described in the initial stages of dental biology research.

We are currently generating transgenic mice with dental epithelium/mesenchymal specific deletions of Gpx4. This will allow us to further investigate the developmental defects related to ferroptosis and ferroptotic stress.